# Long-Term Safety of Growth Hormone Deficiency Treatment in Cancer and Sellar Tumors Adult Survivors: Is There a Role of GH Therapy on the Neoplastic Risk?

**DOI:** 10.3390/jcm12020662

**Published:** 2023-01-13

**Authors:** Carolina Di Somma, Elisabetta Scarano, Rossana Arianna, Fiammetta Romano, Mariarosaria Lavorgna, Domenico Serpico, Annamaria Colao

**Affiliations:** 1Endocrinology, Diabetes and Andrology Unit, Department of Clinical Medicine and Surgery, University of Naples “Federico II”, 80131 Naples, Italy; 2UNESCO Chair “Education for Health and Sustainable Development”, University of Naples “Federico II”, 80131 Naples, Italy

**Keywords:** GH treatment, cancer risk, GHD

## Abstract

Experimental studies support the hypothesis that GH/IGF-1 status may influence neoplastic tissue growth. Epidemiological studies suggest a link between GH/IGF-1 status and cancer risk. However, several studies regarding GH replacement safety in childhood cancer survivors do not show a prevalence excess of de novo cancers, and several reports on children and adults treated with GH have not shown an increase in observed cancer risk in these patients. The aim of this review is to provide an at-a-glance overview and the state of the art of long-term effects of GH replacement on neoplastic risk in adults with growth hormone deficiency who have survived cancer and sellar tumors.

## 1. Introduction

Since t mass manufacture began in 1985, the biosynthetic growth hormone (GH) has been used to treat both growth hormone deficiency (GHD) in adults and a variety of disorders in children linked to short stature, with different approved purposes in different nations [1]. Due to the development of GHD associated with the malignancy and/or severe effects of its treatment, including chemotherapy, surgery, radiation, and biological therapy (antigen–specific monoclonal antibodies and cytotoxic T-cells), certain cancer survivors may need to be treated with GH. Over the past forty years, there has been a significant improvement in cancer treatment, leading to a significant rise in the number of cancer survivors. Childhood cancer treatment frequently results in poor longitudinal growth and GHD [2]. GH replacement therapy improves symptoms and comorbidities linked to GHD in adults. This review summarizes the most recent medical literature regarding GH treatment safety in cancer and sellar tumor adult survivors, with a focus on key issues: Where clinical practice lacks clinical practice consensus.

## 2. Cancer Incidence in Adult Patients

Cancer incidence and mortality are both rising fast globally. There are many causes, but they take into account population aging and growth, as well as shifts in the prevalence and distribution of major cancer risk factors, many of which are linked to socioeconomic development. According to previous estimates, there were 9.6 million cancer deaths and 18.1 million new cancer cases in 2018 [3]. Lung cancer, which affects both sexes equally, is the most frequently diagnosed cancer (11.6% of all cases) and the most fatal (18.4% of all cancer deaths). It is closely followed by female breast cancer (11.6%), colorectal cancer (10.2%), and prostate cancer (7.1%) in terms of incidence and colorectal cancer (9.2%), stomach cancer (8.2%), and liver cancer (8.2%) in terms of mortality [4].

## 3. Cancer Risk in Adult Hypopituitary Patients

While the issue of malignancy is still debatable, it is agreed that adult GHD is linked to an elevated mortality rate from cardiovascular and cerebrovascular disorders in an untreated state. Retrospective investigations have shown contradictory findings regarding the prevalence of cancer among adult GHD patients [5]. Malignancies were found to be less common in hypopituitary men who were not given GH, although they were more common in hypopituitary women who were not given GH [6]. Other studies [7,8] indicated that hypopituitary patients without GH replacement therapy had a higher rate of cancer. When compared to the general population, patients with non-functioning pituitary adenomas (NFPAs) had a 3.91-fold higher prevalence of malignant tumors [9]. Table 1 reports the main studies available in the literature concerning the risk of neoplasia and GH therapy.

## 4. Background on GH/IGF-I Axis and Cancer

Since the early years of the 20th century, researchers have recognized that GH has effects on mitosis, cell differentiation, and growth. It was depicted in the 1950s that peripheral tissues could operate as a conduit for GH action. Subsequently, functions of insulin-like growth factor I (IGF-I) in both healthy and unhealthy cell growth and metabolism have been gradually uncovered [27,28]. It has been demonstrated in experimental animals that endocrine and paracrine actions of GH and IGF-I are involved in cell division and distinction, angiogenesis, and apoptosis suppression, both directly and through interaction with other growth factors [29,30,31,32]. GH plays a role in the multiple stages of carcinogenesis and IGF could possibly increase the incidence of mutations by speeding up the process of DNA repair during the fast development of cancerous cells [33]. On the other hand, other players in the system, such as IGF binding protein 3 (IGFBP3), the GH-IGF system, and IGF-II receptors, prevent the induction of apoptosis, promote mitogenesis, and control IGF-I activities, which operate as tumor growth inhibitors [29,30,31,34,35,36] (Figure 1). Moreover, various hematologic and solid cancers have been linked to local GH, IGF-I, and IGFBP synthesis, the typical or irregular expression of a number of GH-IGF receptor systems, and GH- and IGF-I-induced miRNA dysregulation [29]. A thorough analysis of the potential contribution of the GH-IGF-I system to the emergence of particular types of cancers has been conducted in a number of recent articles [29,31]. Epidemiological study indicates a link between blood IGF-1 levels and the incidence of prostate, breast, and colon cancers [37], raising questions about the safety of long-term GH replacement in the event of de novo neoplasia, tumor regrowth, or tumor recurrence. It is important to know that some intracellular signaling pathways, identified as the third most highly related to breast cancer susceptibility among 421 pathways containing 3962 genes in a human-genome-wide association study, are stimulated by GH [38]. In breast cancer, local expression of GH seems to affect the epithelial–mesenchymal transition in cancer cells, whereas pituitary and exogenous GH are less involved [39,40]. Additionally, a theory to explain the growth of neoplastic colon tumors has been proposed, according to which excessive levels of endocrine or autocrine GH, such as those caused by acromegaly or injury and inflammation of colonic DNA, inactivate genes involved in tumor suppression, inhibit apoptosis, and accelerate the epithelial-to-mesenchymal transition, resulting in alterations that promote malignancy in the intestinal mucosal field modification [41]. As a result, various elements of investigations into the signaling cascades of the GH-IGF-I axis have been indicated as potential therapeutic goals for colon and breast cancer and other cancer types [42,43,44]. Data gathered from genetically or naturally occurring sources of modified animals displaying regular or irregular GH production or action have also supported a link between the GH axis and carcinogenesis [43,45]. Suppression in several of these models of the GH-IGF signaling system has been linked to significantly lower cancer incidence rates and a lengthened life. However, transgenic mice with high amounts of GH in the blood or IGF-I tissue overexpression demonstrate an elevated tumor hyperplasia development risk [29,46].

## 5. GH Treatment in Adult Cancer Survivors

Many studies [48,49] have shown that in adults with pituitary deficiencies and confirmed GHD, there are distinctive characteristics. These include being overweight and abdominally obese, having a lower lean mass, less extracellular water, dyslipidemia (high cholesterol and triglycerides), and low bone mineral density (BMD), which is associated with a major risk of vertebral fractures [49]. These patients also report a lack of energy and well-being, possibly as a result of having less ability for activity and muscle strength. Over the past 20 years, expert panels’ support for GH replacement has grown as more and more of these outcomes have been demonstrated to improve with treatment [49,50,51,52]. Adults with various hypothalamic-pituitary abnormalities, including GHD, have also been documented to have an increased mortality rate from cardiovascular risk factors. The particular role of GHD in mortality in these patients is still unknown, especially as cancer or craniopharyngioma survivors had significantly greater mortality rates than those with NFPA [53,54].

The increased risk of tumor recurrence and the potential association with secondary malignancies may explain the reluctance to test GHD and subsequent GH replacement therapy for at-risk adult childhood cancer survivors (CCS) [5]. More than 99% of patients with GHD have not received GH replacement therapy, as described in the St. Jude Lifetime Cohort study [55]. GHD and hypopituitarism were frequently documented consequences of CCS, particularly in adult patients who received treatment for childhood leukemia in the 1970s [55,56] and those who had survived central nervous system (CNS) malignancies. During the 1990s, a smaller percentage of people were receiving radiation therapy for these malignancies [56]. In adult cancer survivors, generally, the same diagnostic strategy for hypothalamic-pituitary disorders as patients with disorders caused by other conditions can be used [50,57]. Therefore, the Insulin Tolerance Test (ITT) is considered the ‘gold standard’, but since this test has many contraindications (for example, patients with a history of seizures or ischemic heart disease), dynamic tests with glucagon, clonidine, arginine, or arginine with the growth-hormone-releasing hormone (GHRH) are good alternatives [50,57]. We need to pay attention to patients who received brain radiation or other treatments that have caused hypothalamic damage, in which GHRH and combined GHRH-arginine stimulation tests may not be decisive [57]. A safe and well-tolerated oral ghrelin mimic, macimorelin, demonstrated diagnostic precision comparable to the ITT [58]. Unless IGF-I is also under the reference range lower limit, it is suggested to perform one stimulation test to identify GHD in people without other pituitary hormonal abnormalities [49]. The clinical suspicion of GHD associated with certain risk factors such as low IGF-I levels, younger age, a longer time before starting therapy, and variables linked to the site of the tumor and its treatment, suggest the need to test GHD [57]. In order to achieve an appropriate clinical response, such as a decrease in the Adult Growth Hormone Deficiency Assessment (AGHDA) score, in adult cancer survivors on GH treatment, it is recommended to individualize dose titration in order to gain IGF-I levels within the reference range related to age [50,51,59]. IGF-I is known to respond less to GH therapy in women on oral estrogen, therefore in hypopituitary women treated with GH, it might be useful to have non-oral estrogen replacement therapy [60]. The benefits that can be anticipated from GH therapy are similar to those reported in patients with GHD secondary to other causes not related to cancer, including improvements in cholesterol and triglycerides, muscle strength, systolic function of the left ventricular, quality of life (QoL), and cognitive function [61,62]. A meta-analysis reported that the risk of cancer could be lowered [63], but this evidence is likely due to selection bias. In hypopituitary adults on GH treatment, cancer risk was observed to be similar compared to individuals who were not under GH therapy [21,64,65]. Data from the HypoCCS study showed no evidence of an increase in primary cancer with GH replacement in hypopituitary adults compared with the general population [21]. Over the course of ten years of actual clinical practice, data from the ongoing PATRO pharmaceutical-sponsored post-marketing surveillance research on children (*n* = 136) and adults (*n* = 293) with GHD were evaluated. Compared to other GH therapies, there was no elevated incidence of neoplasia [66]. Additional analyses from the PATRO database with 1293 adults, 637 of which (49.3%) were GH-treatment-naive at baseline and the majority of who were deficient in multiple pituitary hormones (*n* = 1128, 87.2%), showed that GH treatment did not increase the risk of cancer. However, it was not possible to evaluate the second cancer risk in patients with prior cancer. On the other hand, according to the SAGhE investigations, young adults with childhood-onset GHD who underwent GH replacement had an elevated risk of several cancers or an increased risk of mortality trend [67]. Divergent sampling sizes and study designs, different kinds of cancer or their prevalence in controls, associated pituitary disease, surgery, irradiation, concomitant comorbidities, and suboptimal pituitary hormone replacement are just a few confounding variables that could account for these discrepant results. Therefore, when considering GH replacement medication, doctors must traverse difficult safety considerations, particularly in newly diagnosed GHD cancer survivor adults and in patients who were under GH therapy and then developed cancer.

The Endocrine Society currently advises waiting one year of disease remission after treatment in childhood cancer patients and then starting GH medication [68]; however, it is unclear if this is sufficient or if waiting longer is safer given the lack of studies that have explored this. The decision of whether to begin GH treatment or not in the case of chronic or non-eradicable cancer should be tailored based on the kind of cancer and the patient’s comorbidities. After a full conversation with the patient and approval from the oncologist, the decision regarding GH therapy should be made on an individual basis [68,69]. The label has a “black box” warning against using GH in patients with active cancer, and the published guidelines do not advise treatment. In the case of a strong desire of the patient to start GH treatment, it could be possible to consider therapy at least one year following remission of a benign hypothalamic-pituitary tumor or up to five years after remission of solid tumors such as breast cancer [57]; nevertheless, it is important to advise patients that there are no definitive data on the link between GH therapy and cancer risk. The advantages of GH therapy should be closely weighed concerning the potential yet unproven higher risk of malignancy.

It is necessary to conduct a prospective surveillance study of a sizable cohort of patients under GH treatment and untreated GHD patients with a history of cancer remission, but this will be difficult given the high cost and the adaptation required for the demographic differences between GH-treated and untreated subjects [70]. In the study of Hartman and coll. [70], using the HypoCCs database, the results showed no association between GH replacement and cancer when comparing GH-treated with untreated patients, despite the fact that GH use is contraindicated in the presence of active malignancy.

Daily subcutaneous GH injection compliance is frequently difficult. The majority of children and adults fail to take their daily GH injections, according to many studies [71,72], which results in high treatment dropout rates [73,74]. In order to reduce injection frequency and increase treatment adherence, efforts have been made to develop long-acting GH formulations [75,76,77]. Two long-acting GH formulations are currently on the market in Asia, several others are in the last stages of development, and one has already received approval in the United States (US) and European Union (EU) but is not yet on the market [75,76,77]. According to recent evidence, long-acting GH medications are safe for non-oncological patients with GHD [75,76,77]. However, it could be claimed that the duration of these studies is insufficient to answer this question. Further long-term studies are necessary to determine the safety of long-acting formulations in GH-deficient patients who survived cancer, taking into consideration the differences in pharmacokinetic and pharmacodynamic profiles when compared to daily treatment.

### 5.1. GH Treatment and Second Neoplasm Risk

During cancer survivor follow-up, the assessment of the risk of a second neoplasm plays an important role, as CCSs are significantly more likely to develop one during their life. The second malignancies that have been most frequently found are non-melanoma skin cancer, soft tissue sarcomas, breast cancer, thyroid cancer, meningiomas, and CNS tumors [78,79]. The risk among survivors could be increased by genetic factors working together with side effects from cancer treatment [55]. Cranial radiotherapy has been associated with a higher risk of developing meningioma [56,80]. While there was no statistically significant link found between GH replacement therapy and primary cancer recurrence, studies have found that GH therapy may enhance secondary neoplasia risk in CCS [14,15,23,79,81,82]. Of the 10,403 patients on GH treatment in the SAGhE cohort, 38 had meningioma. Thirty of them received cranio-spinal radiotherapy, and the overall cohort had a standardized incidence ratio (SIR) for meningioma of 75.4 [24]. However, the risk of meningioma was not increased in patients with no previous cancer diagnosis (SIR = 2.4) and was not significantly associated with the GH therapy start age, time since starting therapy, mean daily GH dose, treatment length, or total GH dose [24]. Sklar and colleagues [15] found a higher risk of second solid cancers after starting GH (RR, 3.21), primarily in patients who survived acute leukemia/lymphoma (RR, 4.98). They did not find secondary recurrence [15]. In a 14.5-year follow-up study on patients treated with CNS irradiation, there were no significant differences in recurrent or secondary tumors between GH-treated and controls [18]. During a mean latency time of 22.5 years, both GH-treated and untreated patients had a 10% risk of GH-dependent effects on a secondary tumor [18]. In groups under GH therapy, meningiomas were the most frequent secondary tumors [16,19]. Meningioma development was associated with sex (major in female), younger age at first cancer diagnosis, longer time after brain radiation [2], a radiation dose of 20 Gy on CNS, and total doses of alkylating agents [83]. Therefore, the presence of several confounding factors, such as radiation therapy, makes a real impact of GH treatment on second neoplasm risk difficult to understand [57]. These findings emphasize the importance of approaching the secondary tumor risk in patients on GH treatment with caution. It is recommended that the slight number of events, high confidence intervals, and lower significance levels must be considered. Furthermore, there is a risk of outcome bias because the secondary neoplasm risk is already increased in CCS [23].

### 5.2. Growth Hormone Therapy and Sellar Tumors

#### 5.2.1. Craniopharyngioma

Craniopharyngioma accounts for approximately 5–10% of pediatric CNS tumors and 3% of CNS tumors in all ages, with major incidences between the ages of 5 and 9 and 40 to 44. At the time of diagnosis, in 40–87% of children, we find at least one hypothalamic-pituitary hormone deficiency [84,85,86]. Craniopharyngioma is treated primarily surgically, and then adjuvant radiotherapy may be useful [84,85,86]. In patients who have, at the time of craniopharyngioma diagnosis, normal levels of GH after a dynamic test, GHD can frequently develop after surgery or subsequent radiotherapy [86]. Progression rates after surgery alone range from 71–90% to 21% when radiotherapy is used after resection [85]. The potential effects of GH on patients with craniopharyngioma are the source of safety concerns about GH treatment.

The majority of information on this tumor comes from case-control studies. Most of these studies were conducted in patients with both childhood and adult-onset disease. A study conducted by Olsson et al. [87] in all age groups, reported residual tumors in 29% of patients on GH treatment, whereas 47% of patients in the untreated group had residual tumors. Up to 15 years after treatment, the two groups exhibited no differences in tumor progression. Other studies regarding patients with craniopharyngiomas alone or other sellar tumors including craniopharyngiomas did not find evidence of craniopharyngioma recurrence in patients under GH treatment [10,88,89,90,91]. Moreover, no increased risk of craniopharyngioma relapse after neurosurgery was found in patients on GH treatment in a recent, single-center, retrospective study involving 89 patients with adult-onset craniopharyngioma with a median treatment follow-up of more than 7 years [92].

Pharmaceutical-sponsored post-marketing surveillance studies, conducted with patients with craniopharyngioma during childhood and adulthood, have also yielded safety data. The KIMS (Pharmacia & Upjohn International Metabolic Database) report found craniopharyngioma in 12% of 1000 adult patients present on the registry at the time of analysis. Only six patients with craniopharyngioma were reported to have tumor relapse on GH treatment [93]. With a mean follow-up of 4.8 years, the HypoCCS study of 1058 adults with craniopharyngioma found no association of GH treatment with the risk of recurrence (RR 1.32; range 0.53–3.31, *p* = 0.55) [21]. Other open-label post-marketing surveillance registries have also evaluated the risk of relapse in children with craniopharyngiomas on GH treatment and they found no increased recurrence of the tumor [81,94,95]. A recent meta-analysis of ten studies on GH replacement in childhood-onset craniopharyngiomas patients compared 3436 GH-treated to 51 GH-untreated children. They found a lower recurrence risk of craniopharyngioma in children under treatment, though this could be due to selection bias in individual studies preferring to start GH therapy in patients with less aggressive tumors [96].

The risk of metabolic complications also increased. In the KIMS study, patients with craniopharyngioma were nine times more likely to develop diabetes mellitus compared to the general Swedish population [97]. Nevertheless, the risk of diabetes mellitus in this population was not increased by GH [98], though, in one study, a reduction in insulin sensitivity was found after a longer period of GH therapy [99].

#### 5.2.2. Pituitary Adenomas

NFPA accounts for most pituitary lesions that cause GHD in adult patients. In the major GH replacement therapy registries, NFPAs and craniopharyngiomas are the most frequent tumors [100,101]. The data on the progression or relapse of NFPAs under GH treatment are encouraging. No difference in NFPA relapse was found between 74 patients GH-treated and 120 patients GH-untreated in a recent, large, single-center study of adult patients with NFPA who underwent surgery [92]. In a 2009 retrospective case-control study on approximately 200 patients with NFPA and GHD providing data with tumor-free progression over 10 years, no difference in NFPA progression was found in GH-treated patients compared to untreated [102]. These reassuring findings are consistent with other well-conducted case-controlled studies and post-marketing surveillance studies [21,93,103,104,105,106] that included up to 14 years of follow-up.

In patients with acromegaly (acroGHD), contrasting data were reported about GH safety in the management of GHD. Over 6 months, in a randomized placebo-controlled trial of thirty patients who underwent GH replacement [107], an improvement in body composition and QoL was found with no adverse events or safety problems, despite an earlier open-label study conducted on twenty patients that found increased cardiovascular risk in patients treated with GH [108]. A retrospective analysis conducted in acroGHD and NFPA GHD patients from the KIMS dataset, which were treated with GH and compared to a background reference population, revealed higher cardiovascular mortality in the acroGHD group compared to the general population and the NFPA group. It was not possible to understand whether this was dependent on GH treatment or on prior acromegaly, but in acroGHD, replacement therapy with GH should be considered with caution. Notably, markers of glucose tolerance decreased over time in both the acroGHD and NFPA groups treated with GH [109]. Moreover, some case-control studies show no increased risk of recurrence or progression of NFPAs and craniopharyngiomas with GH therapy, including those patients who have post-operative tumor remnants and those patients treated with or without radiotherapy [87,90,92]. Although the data show that GH therapy does not increase the overall risk of recurrence/regrowth of the original tumor, the majority of craniopharyngiomas and NFPAs that recur will recur within 7 and 5 years, respectively, with growth becoming less common after 10 years [110]. Many studies conducted on the safety of GH treatment in patients with sellar tumors have shorter follow-up periods, which may undervalue recurrence and regrowth risk.

## 6. Conclusions

Outcomes for long-term GH replacement safety in adolescent and adult patients with GHD confirmed a favorable safety profile of GH, despite the evidence of a link between the GH/IGF-I axis and cancer risk supported by some epidemiological and experimental studies. In cancer survivors, studies conducted on data from international databases reported an increased risk of de novo malignancies in younger patients and a decreased risk in those with idiopathic/congenital GHD. Moreover, there was no evidence of an increase in primary cancer with GH replacement in hypopituitary adults compared with the general population and no association between GH replacement and cancer when comparing GH-treated and untreated patients. Some other factors seemed to contribute to neoplastic risks, such as sex, younger age at first diagnosis of cancer, a longer time period after cranial irradiation, cumulative radiation dose, and cumulative doses of other chemotherapeutic agents.

In conclusion, long-term cancer surveillance is still required for all rhGH-treated patients, especially for those who have factors that increase their chance of developing cancer.

## Figures and Tables

**Figure 1 jcm-12-00662-f001:**
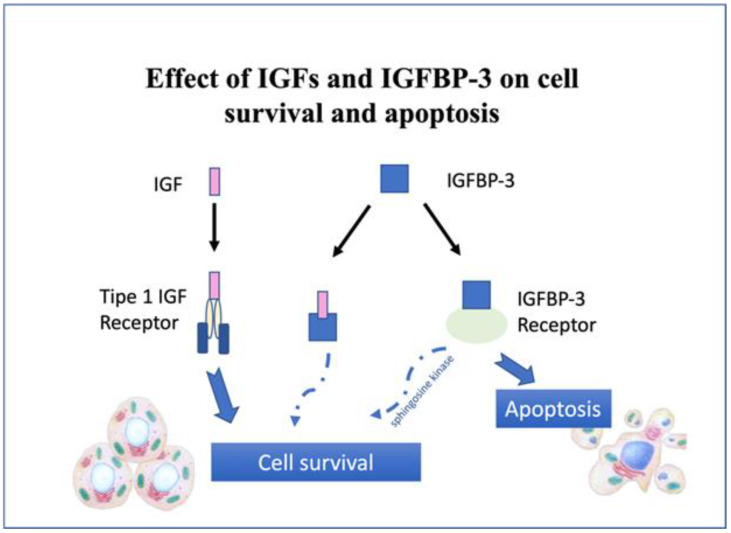
IGFBP-3’s and IGF’s role in Apoptosis and Cell Survival: IGFBP-3 plays double function of gate-keeper (induction of apoptosis and cell cycle arrest) and care-taker (DNA repair) through mechanisms independent of IGFs [47].

**Table 1 jcm-12-00662-t001:** Studies reporting data on cancer risk and GH therapy.

Reference	Study Group	New Malignancy/Recurrence		Conclusions
Arslanian et al., 1985 [10]	34 CNS tumors: Germinomas (4), craniopharyngiomas (18), astrocytomas (3), medulloblastomas (2), others (7); 94% GHD	**GH-treated (24/34):** 8 (33%) recurrences. Follow-up 8–72 months after GH therapy initiation	**Non-GH treated (10/34):** 3 (30%) recurrences	Likely no association between GH therapy and tumor recurrence
Clayton et al., 1987 [11]	31 patients with cranial irradiation for medulloblastoma (14), gliomas (8), ependymomas (2), leukemia (6), lymphoma (1); all GHD	**GH-treated (21/21):** 5 (23,8%) recurrences (1 optic nerve glioma, 2 medulloblastomas, 2 ependymomas); 3 during and 2 after GH treatment		Risk of recurrence of medulloblastoma, glioma, and leukemia did not increase
Corrias et al., 1997 [12]	Patients irradiated for brain tumors with GHD: 25 GH-treated (11 medulloblastomas, 8 gliomas, 6 ependymomas); Control group: 100 non-GH treated	**GH-treated:** 4 tumor recurrences (16%)	**Non-GH treated:** 18 tumor recurrences (18%)	Risk of brain tumor relapse after radiotherapy and GH therapy did not increase
Nishi et al., 1999 [13]	Japanese cohort of 32,000 GHD patients on treatment from 1975 to 1997	**GH-treated:** 14 leukemia and 1 myelodysplastic syndrome (6 with risk factors). 9 observed in patients without risk factors. Cases expected: 6.96–9.28		In patients on treatment with GH without risk factors, the incidence of leukemia did not increase
Leung et al., 2002 [14]	47 CCS after ALL with GHD and treated with GH for 1 to 8 years. Control group: 544 CCS after ALL, non-GH treated	**GH-treated:** 1 case of sclerosing sweat duct carcinoma of the scalp, 1 case of myelodysplastic syndrome, no cases of leukemia recurrence	**Non-GH treated:** 8 leukemia relapses, 16 s tumors	In CCS after ALL with GHD, GH replacement is safe
Sklar et al., 2002 [15]	CCS: 361 treated with GH for 4.6 years. 12,963 non-GH-treated. Follow-up: 6.2 years.	**GH-treated:** 9 recurrences; 15 s tumors (all solid tumors, no leukemia)	**Non-GH treated:** 502 recurrences, 344 s tumor	In CCS GH-treated patients, there was no increased risk of cancer recurrence or death, but secondary solid tumor risk increased
Ergun-Longmire et al., 2006 [16]	CCS patients: 361 GH-treated for 4.6 years (0.1–14). 13,747 non-GH-treated. Follow-up: 5 years	**GH-treated:** 20 s tumors (9 meningiomas)	**Non-GH treated**: 555 s tumors (62 meningiomas)	In CCS-treated patients with GH, there was an increased risk of second solid tumor, but this risk seems to decrease with a longer follow-up
Wilton et al., 2010 [17]	KIGS (Pfizer International Growth Database): 58,603 patients (54% IGHD, 11% TS, 7% SGA, others). Mean follow-up: 3.6 years.	**GH-treated:** 32 new malignancies (9 CNS tumors, 3 NHL, 3 leukemias, 3 testicular cancers, 14 others)		There was a similar incidence of cancer between young GH-treated patients without known risk factors and general population
Mackenzie et al., 2011 [18]	Brain-irradiated CCS: 110 GH-treated for 8 years, follow-up of 14.5 years. 110 non-GH-treated, follow-up of 15 years.	**GH treated:** 6 tumor recurrence, 5 s tumors (4 meningiomas)	**Non-GH treated**: 8 tumor recurrence, 3 s tumors (2 meningiomas)	risk for recurrence or second neoplasm did not increase
Patterson et al. 2014 [19]	Childhood Cancer Survivor Study: 12,098 CCS, 338 GH-treated, 11,760 non-GH-treated. Follow up 15 years	**GH-treated:** 16 (4.7%) second tumors (10 meningiomas, 6 gliomas), 8 (2.4%) recurrences	**Non-GH-treated:** 203 (1.7%) CNS tumors (49 gliomas, 138 meningiomas, 16 others), 178 (1.5%) recurrences	After GH therapy, there was no increased risk of CNS second tumor. Meningiomas are more frequent, and the increased risk of glioma was linked to high dose of cranial radiation
Brignardello et al., 2015 [20]	49 GHD CCS patients (34 brain tumors, 10 ALL, 5 AML), 45 with cranial irradiation. 26 GH-treated; 23 non-GH-treated. Median follow-up: 16 years	**GH-treated:** 8 s tumors (5 meningiomas)	**Non-GH-treated:** 6 s tumors (4 meningiomas). No second tumor in 4 GHD patients without radiotherapy	Secondary tumor risk did not increase in CCS treated with GH. The most important risk factor for development of second tumor was radiotherapy.
Child et al., 2015 [21]	HypoCCS database: 8418 patients GH-treated, 1268 patients untreated. Of these, 3668 GH-treated and 720 untreated patients had PA, and 956 GH-treated and 102 untreated patients had CP.	**GH-treated:** 225 (breast, prostate, colorectal cancer)	**Non GH-treated:** 45 (breast, prostate, colorectal cancer)	No significant difference was observed in the incidence of primary malignancies between GH-treated and untreated patients.
Libruder et al., 2016 [22]	GH-treated patients: 1687 low-risk (IGHD, SGA; follow-up 6.5 ± 4 years) and 440 intermediary-risk (MPHD, TS, SPW; follow-up 8.1 ± 4.6 years).	**GH-treated:** Low-risk group: 2 cases of malignancy, SIR 0.76; Intermediary-risk group: 4 cases of malignancy, SIR 4.52		Cancer risk was increased only in the intermediary-risk group and not in the low-risk group.
Swerdlow et al., 2017 [23]	SAGhE (Safety and Appropriateness of Growth Hormone Treatments in Europe): 23,984 GH treated patients: 52% isolated growth failure (GHD, ISS, SGA), 14.6% TS, 10.4% MPHD, 9.3% CNS tumor, others.	**GH-treated:** In GH-treated patients with no previous cancer, SIR was 2.8 for bone and 16.3 for bladder. GH dose was not related to cancer risk, but for patients treated after previous cancer, the risk of cancer mortality increased with increasing daily dose		No increased risk in patients with isolated growth deficiency. Bone and bladder cancers increased in non-CCS group. Increasing daily GH-dose in CCS was related to increased cancer mortality risk.
Swerdlow et al., 2019 [24]	SAGhE (Safety and Appropriateness of Growth Hormone Treatments in Europe): 10,403 GH treated patients: 38% isolated growth failure (GHD, ISS, SGA), 16.5% TS, 12.9% organic MPHD, 12.6% CNS tumor, others. Mean follow-up: 14.9 years.	**GH-treated:** Non-cancer group: 1 meningioma (TS), SIR: 2.4; CCS (*n* = 1830): 37 meningiomas, SIR 466.3. CCS with previous radiotherapy (*n* = 1178): 30 meningiomas, SIR 658.4.		In patients treated during childhood with GH, there was a very high relative risk of meningioma after cranial radiotherapy for malignancy. The increased risk is primarily due to radiotherapy. Mean daily or cumulative dose and duration of GH therapy were not related to cancer risk.
Child et al., 2019 [25]	GeNeSIS (Genetics and Neuroendocrinology of Short Stature International Study): 22,311 GH-treated children: 58% GHD, 4.8% GHD after CNS tumor, 13% ISS, 8% TS, 6% SGA; 20,556 no previous cancer, 622 previous cancer. Follow-up 4.2–3.2 years; 456 non-GH treated (192 no previous cancer, 114 previous cancer)	**GH-treated:** 85 recurrences in 74 of 1087 (6.8%) GH-treated children with a history of previous neoplasm (77 intracranial tumors). In patients without cancer history, 14 primary cancers were observed [SIR: 0.71 (0.39, 1.20)].	**Non-GH-treated:** 9 recurrences in 9 of 148 untreated patients (6.1%)	No increased risk for primary cancers
Johannsson et al., 2022 [26]	KIMS study: 15,809 GH-treated patients of whom 14,533 did not have a prior history of cancer	**GH-treated:** 471 primary cancers in patients without cancer history. Prostate (*n* = 86), nonmelanoma skin (*n* = 57), breast (*n* = 39), lung (*n* = 37), brain (*n* = 29), melanoma (*n* = 25), and colon (*n* = 20) cancers. Second cancer developed in 3.2% (15/471) of patients		The overall risk for all-site de novo cancer among KIMS patients with GHD and no prior history of cancer is comparable to that expected in the general population

ALL, acute lymphoblastic leukemia; AML, acute myeloid leukemia; CCS, childhood cancer survivors; CI, confidence interval; CNS, central nervous system; CP, craniopharyngioma; GH, growth hormone; GHD, growth hormone deficiency; HL, Hodgkin’s lymphoma; IGHD, idiopathic growth hormone deficiency; ISS, idiopathic short stature; MPHD, multiple pituitary hormone deficiency; NHL, non-Hodgkin’s lymphoma; PA, pituitary adenoma; PWS, Prader–Willi syndrome; RR, relative risk; SGA, small for gestational age; SIR, standardized incidence ratio; SMR, standardized mortality; TS, Turner syndrome.

## Data Availability

Not applicable.

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
