# Peer review of "Long-Term Safety of Growth Hormone Deficiency Treatment in Cancer and Sellar Tumors Adult Survivors: Is There a Role of GH Therapy on the Neoplastic Risk?"

_jcm, 2023, doi:10.3390/jcm12020662_

Round 1

Reviewer 1 Report

Reviewer comments on manuscript # jcm-2141158, entitled “Long-term Safety of Growth Hormone Deficiency treatment in Cancer and Sellar Tumors Adult Survivors: Is there a role of GH therapy on the Neoplastic Risk?”

The manuscript is well-written in terms of the best evidence available in this subject that was retrieved from literature.

Some concerns are pointed below.

Comment #1, ALL MANUSCRIPT: although the main body of the manuscript is well-written in terms of grammar, some (but not all) minor improvements may be made, as confusing/awkward sense may be felt by the reader and distract it. Examples are:

Line 239: “In patients with who have normal levels of GH at the time of diagnosis frequently GHD can develop after surgery or subsequent radiotherapy..

Line 245: “in patients which had both childhood and adult-onset disease.

Line 274: “in insulin sensitivity was found after a GH therapy a longer period

Line 276: “NFPA accounts for most pituitary lesions that causes GHD in adult pa-tients.

Line 282: “about 200 patients with NFPA and GHD providing data with free tu-mor

Comment #2, “GH treatment in adult cancer survivors” SECTION: with modifications to the testing strategies, particularly with regard to the Growth hormone–releasing hormone (GHRH) and combined GHRH-arginine stimulation tests after radiation therapy of the brain.” Although I think that the authors may want to say that GHRH test as a stimulation test to prove GH deficiency is not feasible in patients with hypothalamic injury, it is not clear in this sentence only saying “with modifications..”.  Please revise.

Comment #3, “GH treatment in adult cancer survivors” SECTION: “it is suggested to perform two stimulation tests to identify GHD in people without other pituitary hormonal abnormalities”. In adults with suspected GH deficiency, the available consensus guidelines and statements (please refer to: J Clin Endocrinol Metab. 2011; 96(6):1587–1609; Endocrine Practice. 2015;21(7):832-838; Eur J Endocrinol. 2022;21;186(6):P35-P52) do not refer the need to perform two GH stimulation test to prove the clinical suspicion of GH deficiency. If the need to perform two GH stimulation tests is a regulatory obligation for financial approval of GH treatment in the country of the authors, it should then be specified and separated from what are the current recommendations. Please revise.

Comment #4, “GH treatment in adult cancer survivors” SECTION: “In case of strong desire of patients to start GH treatment, it could be possible to consider therapy at least two years following cancer remission”. The authors may consider the recommendations of a new consensus statement recently published on the timing of GH initiation in adult-onset solid tumors (Eur J Endocrinol. 2022;21;186(6):P35-P52).

Comment #5, “Craniopharyngioma” SUB-SECTION: “In patients with who have normal levels of GH at the time of diagnosis frequently GHD can develop after surgery or subsequent radiotherapy”. When the authors state “normal levels of GH” is it in the context of dynamic tests? If not, basal GH levels (exception for newborns with specific clinical signs) are not feasible/sufficient to prove normal GH reserve. Please clarify.

Author Response

Dear Editor and Reviewers,

We made changes to the text of our review according to your instructions and we uploaded the new updated document highlighting the changes made using the "Track changes" function as suggested by the editorial office. Therefore, lines reported in answers to Reviewers’ comments refer to this version of our manuscript.

Thank you very much for your valuable help and attention.

Reviewer 2 Report

Comments to the authors:

It is a good review of the long-term safety of GHD treatment in cancer and sellar tumors adult survivors. The authors addressed all essential topics, as published before, in the consensus statement (2022). However, I have a few comments about it.

1. Introduction: I suggest keeping three paragraphs instead of 2 subsections (1.1 and 1.2). It is optional to understand. In addition, I propose an ultimate paragraph to introduce section 2 of MS (background on GH/IGF1 and cancer).

I need help understanding the Table in the middle of MS. Is it Table 1? Also, what is the Table's title? There is no call for the Table (1 or whatever) in the MS. Still, the authors were talking about cancer and sellar adult survivors and GHD treatment, but I observed many cited references in the Table are about childhood cancer survivors. As I understand, they are two different populations. Data about the risk for primary cancer or cancer recurrence in survivors treated with GH during adulthood are limited. However, we can see data produced for adult patients with benign pituitary adenomas and craniopharyngioma. 

2. Background on GH/IGF1 and cancer:

First, I prefer to use the title "Background on GH/IGF1 axis and cancer" because the GH-IGF axis, IGF2, insulin receptors, and hybrid receptors have essential roles in cancer.

I had to read this section more than twice. It is a long paragraph with a lot of statements and references. The authors should split this long, difficult-to-understand paragraph into 4 or 5 ones. 

Also, this section had similarities with a section with the same title in the consensus statement published by Boguszweski et al. (2022). 

Figure 1: it needed to add more information to understand the document.

3. GH treatment in adult cancer survivors

Line 121-125 - "Despite the potential advantages,..." - is it a personal opinion? The authors cited reference 55 with the experience of a single-center study to justify this statement.

Third paragraph, line 126 - the consensus statement addressed when and which test should be considered—again, summing up this part to improve understanding.

Author Response

(The authors gave the same response as above.)

Reviewer 3 Report

Line 216 grammatical error "no"

Can add more information about colon cancer IGF-II

It will be interesting to compare patients receiving GH replacement with GH naive controls with the dose of RT matched to assess true risk of recurrence.

Also would be interesting to highlight independent effect of GH (without RT) on independent cancers.

Author Response

(The authors gave the same response as above.)

Round 2

Reviewer 2 Report

Thank you for the answers to the questions and the changes made in the MS. I have some minor mistakes to address.

Minor revisions:

line 135: treatment instead of "treat met"

line 178: them instead of "theme"

line 297: start instead of "star"

line 305: patient instead of "pa-tient"

line 311: tumor instead of "tu-mor"